# Long Noncoding RNAs—Crucial Players Organizing the Landscape of the Neuronal Nucleus

**DOI:** 10.3390/ijms22073478

**Published:** 2021-03-27

**Authors:** Hanna Sas-Nowosielska, Adriana Magalska

**Affiliations:** Laboratory of Molecular Bases of Cell Motility, Nencki Institute of Experimental Biology, Polish Academy of Sciences, 02-093 Warsaw, Poland

**Keywords:** lncRNA, neuronal nucleus, nuclear architecture

## Abstract

The ability to regulate chromatin organization is particularly important in neurons, which dynamically respond to external stimuli. Accumulating evidence shows that lncRNAs play important architectural roles in organizing different nuclear domains like inactive chromosome X, splicing speckles, paraspeckles, and Gomafu nuclear bodies. LncRNAs are abundantly expressed in the nervous system where they may play important roles in compartmentalization of the cell nucleus. In this review we will describe the architectural role of lncRNAs in the nuclei of neuronal cells.

## 1. Introduction

The cell nucleus is a highly organized organelle composed of many functional and structural domains (Figure 1). Silent chromatin is compacted in the form of dense chromocenters, localized at the nuclear periphery or adjacent to the nucleolus [1]. The active chromatin compartment faces the interchromatin domain in which nuclear bodies reside [2,3]. Nuclear bodies are membrane-less nuclear subcompartments that form via liquid–liquid phase separation. They are formed by various RNAs and proteins, which together conduct highly specialized processes. Examples include the nucleolus, speckles, paraspeckles, and Promyelocytic leukemia (PML) bodies [4], just to mention a few [5]. Importantly, nuclear architecture is dynamic and directly linked to transcriptional activity [6,7,8].

Neuronal cells might be of particular interest when it comes to analyzing nuclear architecture and organization, as different types of neurons differ significantly in terms of nuclear size and chromatin organization [9]. They dynamically respond to the external stimuli with a transcriptional burst, and some evidence suggests that neuronal activation is connected with reorganization of the nuclear structure [10,11,12,13,14,15]. Moreover, neurons seem to differ from other cell types in the composition and organization of at least some of the nuclear bodies [4,16,17].

Recently, long non-coding RNAs (lncRNAs) were described as potent organizers of nuclear architecture [18,19]. The nervous system is particularly enriched in lncRNAs. In primates almost 40% of discovered lncRNAs are expressed specifically in the brain [20]. Compared to protein-coding genes, these non-coding RNAs show lower expression levels, but their expression is more specific in terms of cell type, tissue, and spatiotemporal pattern [21,22,23]. Importantly, several studies show that dysregulation of lncRNA expression is correlated with aging and neurodevelopmental and cognitive disorders like autism [24,25,26], Rett syndrome [27], Huntington’s disease [28], or Alzheimer’s disease [29].

LncRNAs can affect nuclear organization by several mechanisms. They may act as epigenetic modulators of chromatin states, thus affecting local chromatin accessibility. Numerous lncRNAs have been shown to interact with epigenetic modifiers like polycomb proteins or methyltransferases and recruit them to particular gene loci [31]. In turn, changes in epigenetic marks affect local chromatin condensation levels [32,33]. One interesting example of such a mechanism is Meg3 and other lncRNAs derived from the Dlk1-Dio3 locus. In motor neurons, these lncRNAs have been found to maintain the repressed state of HOX genes, most likely by guiding polycomb repressive complex (PRC) proteins to this locus [34]. LncRNA can also determine the condensation state of large chromatin domains. One of the best characterized examples is the inactivation of the whole X chromosome, which is orchestrated by *XIST* lncRNA (described below) [35]. Additionally, lncRNAs, like *MALAT1*, can reshape the local nuclear landscape by redistributing particular chromatin domains to active or silent nuclear compartments [36]. Moreover, acting as a molecular scaffold, lncRNA like Gomafu, *NEAT1*, and *MALAT1* recruit proteins to form nuclear bodies like Gomafu bodies, paraspeckles, and speckles, respectively [37,38,39,40]. All of these mechanisms might be of particular interest when analyzing changes in nuclear architecture in neuronal cells upon stimulation. Much focus has been put on the regulatory role of lncRNAs in neurons, while a comprehensive review of their structural role is lacking. Here we will focus on lncRNAs, which directly affect the organization of the neuronal nucleus, in particular by participating in nuclear body formation.

## 2. LncRNAs Involved in Nuclear Body Formation

### 2.1. XIST and Barr Body

#### 2.1.1. The X Chromosome Inactivation

In 1949, Barr and Berttram [41] described a nuclear body present in neurons of a female cat, which was localized adjacent to the already known nucleolus. The authors suggested that the newly discovered structure, called a nucleolar satellite, was somehow connected with sex chromosomes, as it was virtually nonexistent in neurons derived from male cats. Later studies by Ohno et al. [42] and Lyon [43] showed that this nuclear assembly comprises heterochromatin of one of the X chromosomes.

Each female cell has only one active X chromosome (Xa), where the second one is transcriptionally inactivated during embryonic development and forms the Barr body, named after its finder. The paternal and maternal copies are randomly silenced in cells of females, which results in the haphazard distribution of actively transcribed genes inherited from both parents on X chromosomes. The inactivation of one copy of chromosome X of females is necessary to balance gene expression quantity in males and females. Moreover, in 1967, Susumu Ohno proposed that the monoallelic expression of X-linked genes in mammals should be compensated by the twofold transcriptional up-regulation of the active X in both males and females [44]. However, the recent analysis of 500 public RNA-seq datasets from multiple tissues and species showed that in mammals and birds, the ratio of global transcription of sex chromosomes to autosomes is approximately 0.5, which is inconsistent with Ohno’s assumption. In insects, fishes, and flatworms, this ratio is close to 1 [45].

The full dosage compensation is achieved epigenetically on many levels and was first described for *Drosophila* sex chromosomes by Muller [46]. In fruit flies, contrary to mammals, it relies on the upregulation of the male X chromosome. Here the central role is played by two long non-coding RNAs, *roX1* and *roX2,* which together with five proteins (MSL1, MSL2, MSL3, MLE, and MOF) form the male-specific lethal (MSL) complex. The two lncRNAs are essential for MSL complex assembly and targeting to the X chromosome, where it stimulates transcription of the X-related genes [47].

X chromosome inactivation (XCI) during development is a dynamic, multi-step, and highly organized process, which is fully dependent on X-inactive specific transcript (*XIST*), a type of lncRNA. It is initiated at the X-chromosome inactivation center (XIC) of the putative inactive X (Xi) chromosome, where the *XIST* gene is localized and expressed [48]. Within the core region of the XIC, no protein-coding genes are present; instead, it contains sequences of several lncRNAs. In addition to XIST, this region harbors genes for *REPA*, *TSIX*, *XITE*, and *JPX*, which are involved in the regulation of *XIST* expression [49]. *TSIX*, transcribed from the antisense strand of *XIST*, is a negative regulator of its expression, and together with *XITE* controls the allelic choice of the active X chromosome [50]. The other two lncRNAs, *REPA* and *JPX*, are shown to be positive regulators of *XIST* [51]. After *XIST* RNA expression and binding at the nucleation center, the silencing spreads throughout the chromosome. Inactivation of the 150 Mb long chromosome is quite a challenge and requires a specific spreading mechanism. Artificial placement of XIC on an autosome results in its silencing induced by the ectopic expression of *XIST* and coating of DNA by the transcript; however, induced inactivation is not as efficient, suggesting some sequence specificity [48,52,53]. Indeed, Gartler and Riggs hypothesized the existence of “way stations”, which would be enriched in the X chromosome and boost the spreading of silencing [54]. The later studies of Boyle and colleagues [55] showed that long interspersed elements 1 (LINE-1) are enriched particularly on X chromosomes, which later allowed Mary Lyon to propose a hypothesis about their involvement in X chromosome inactivation [56]. LINEs belong to autonomous, non-LTR retrotransposons, with 6 to 7 kb long sequences, which constitutes about 17 and 19% of human and mouse genomes, respectively [57]. The LINE-1 density correlates with the efficiency of XCI spreading, being the highest at XIC [58]. Silent repeats tend to cluster in the interphase nuclei and facilitate the assembly of the heterochromatic compartment, which accelerates *XIST* and silencing spreading to the whole Xi [59]. The next steps involve exclusion of RNA polymerase II [60] and removal of histone marks characteristic for active promoters and enhancers [60,61,62] with simultaneous placement of repressive histone tail modifications [62,63,64] and DNA methylation [65,66], which is all leading to chromatin condensation [53]. The inactive X chromosome becomes tethered to the nuclear periphery [67] or/and nucleolus [68,69]. The cascade of these events is mediated by the proteins, which are recruited to inactivated X chromosome through the various functional domains of *XIST* transcript, and in the case of nucleolar localization two other lncRNAs, namely *FIRRE* and *DXZ4* [70,71]. Within the *XIST* sequence, six such domains have been identified. The A–F repeats, which are conserved between humans and mice, overlap with blocks of tandem repeats of unique short sequences, containing motifs specific for RNA binding proteins involved in coordinated silencing of X-linked genes [72].

*XIST* detectable by RNA-FISH colocalizes with a scaffold attachment factor A (SAF-A)/heterogeneous nuclear ribonucleoprotein U (hnRNPU), a nuclear matrix protein [53]. SAF-A/hnRNPU interacts both with DNA and RNA, forms multimers in the presence of nucleic acids [73], and mediates the retention of many RNAs in the nucleus [74]. Its binding to DNA depends on a conserved SAF-box motif, which interacts cooperatively with a minor-groove of multiple clustered A-tracts of scaffold attachment regions (SAR) DNA [75]. Through RGG (arginine–glycine–glycine), RNA-binding domain SAF-A/hnRNPU binds to the D repeat of *XIST* and is indispensable for *XIST* chromosomal localization [74]. Knocking down of SAF-A/hnRNPU as well as deletion of a D-repeat from *XIST* results in the dispersion of *XIST* in the nucleoplasm and lack of Xi silencing and Barr body formation [76]. In turn, the highly conserved repeat A element was shown to be involved both in the regulation of endogenous *XIST* expression [77] and in the triggering of X-linked gene silencing [78]. The latter is dependent on the direct binding of SPEN protein (known also as SHARP-SMRT/HDAC1-associated repressor protein), the depletion of which similarly reduces the inactivation of Xi [79]. However, the presence of A repeat element in *XIST* as well the binding of SPEN is dispensable for *XIST* chromosomal localization in *cis*. SPEN is involved in the recruitment of several members of the repressor complexes like NCoR/SMRT and NuRD, and activation of deacetylation [80]. Loss of histone deacetylase HDAC3 activity was shown to affect Xi silencing [62], as deacetylation is crucial and an early step of X chromosome inactivation. Further steps are mediated by the repressive complexes belonging to the Polycomb group, which bind to *XIST* B and C repeats [63]. Polycomb repressive complex 1 (PRC1) is responsible for the ubiquitination of histone H2A on lysine 119 (H2AK119ub) and recruitment of PRC2, which fully relies on H2AK199ub deposition, whereas PRC2 mediates trimethylation of histone H3 on lysine 27 (H3K27me3), which in turn signals recruitment of PRC1 complexes, potentiating the Polycomb condensate formation [81]. Overall *XIST* recruits over 80 proteins [82,83,84], which regulate multiple and highly organized functions like chromatin remodeling, nuclear matrix binding, and RNA processing, leading to XCI [85].

#### 2.1.2. Xi Structure and Localization

Inactivated X chromosome has a very unusual structure, which was shown in detail using chromosomal conformation capture-based techniques [71,86]. Both mouse and human Xi consist of two megadomains characterized by the frequent intrachromosomal contacts, which are linked by the hinge region where lncRNA coding microsatellite *Dzx4*/*DZX4* repeat is located. The Xa is organized similarly to autosomes into more than 100 topologically associated domains (TADs), on average 1 Mb long, consisting of smaller chromatin loops frequently linking enhancers and promoters. The Xi megadomains are composed of several dozen super loops (between 7 and 74 Mb long) some of which are anchored at regions containing genes of lncRNA: *LOC550643*, *XIST*, *DXZ4*, and *FIRRE*. The unique architecture of Xi depends fully on *XIST*, as inducible *XIST* expression in male mouse cells resulted in increased interaction frequencies along the chromosome, separation of two megadomains, and a final structure similar to female Xi [87]. Interestingly when the mutated form of *XIST* lacking A-repeat region, which is responsible for Xi coating and RNA polymerase II exclusion, was induced, male X chromosome structure was not affected.

The differences in chromatin contacts are reflected in the morphology of chromosome X territories [88,89]. The observed shape of Xi chromosome territory (CT) is more rounded and shows a smooth surface when compared with Xa CT, which has a flatter shape and exhibits a bigger and more irregular surface. Interestingly, the volume of both chromosomes is quite similar. Teller et al. [90] showed the presence of two structural domains in Xi. Authors measured physical 3D distances between gravity centers of FISH probe intensities used as reference points for differentially-labeled segments. This allowed them to estimate chromatin compaction on 30–50 Mb, 10 Mb, and approximately 1–4 Mb length scales. The higher compaction was observed only for segments bigger than 20 Mb. It suggested that the higher condensation in Xi-territory resulted mainly from a regrouping of ~1 Mb chromatin domains rather than from increased compaction within the individual domains. Interestingly, the authors showed that the chromosome territory of the Xi surpass the region of the Barr body characterized by the *XIST* binding and highly condensed chromatin. However, gene-poor and gene-rich as well as expressed and repressed segments were evenly distributed within the whole Xi CT. The super-resolution microscopy showed the details of Xi ultrastructure. It is not a solid, uniformly condensed region of chromatin, but is reminiscent of a sponge-like structure filled with slightly collapsed interchromatin channels. They start at the nuclear pores and meander between areas of the tightly packed chromatin [53]. This again argues for the complete reorganization of chromatin domains within the Xi CT rather than simple condensation leading to higher compaction of chromatin.

As was mentioned, the Xi was observed for the first time in female cat neurons adjacent to the nucleolus [41]. Later studies have shown that regulation of the Xi perinucleolar localization is cell cycle-dependent and relies on *XIST*. The introduction of XIC or *XIST* itself to autosomes is sufficient for their perinucleolar targeting from the mid to late S phase of the cell cycle [68,91]. Moreover, Yang and colleagues have found the involvement of another lncRNA. Knockdown of *Firre* in mouse fibroblasts disrupts perinucleolar Xi localization and H3K27me3 levels. However, upon *Firre* depletion, no reactivation of X-linked genes was observed. Interestingly, in electrically-stimulated motor neurons, the nucleolar satellite observed by Barr and Bertram was moving away from the nucleolus [92]. Similarly, Borden and Manuelidis described the difference in localization of X chromosomes in electro-defined seizure foci of epileptic patients. As compared to the unaffected cells from the neighboring region, both X chromosomes were repositioned towards the nuclear center [93]. Unfortunately, currently there are no data showing how activation induced Xi repositioning influences Xi silencing. Numerous studies showed that both Xa and Xi are preferably localized at the nuclear periphery. Although all mouse and human chromosomes show the presence of lamina-associated domains (LADs) [94], perinuclear localization of Xi is also regulated by *XIST* interaction with membrane attachment nuclear envelope proteins, namely lamin B receptor (LBR), lamina-associated protein 2 (LAP2), and SUN domain-containing protein 2 (SUN2) [67,84]. Binding to nuclear lamina through LBR is indispensable for Xi transcriptional inactivation, as knockdown of LBR leads to defects in the silencing of X-linked genes [67].

In summary, Xi’s unusual ultrastructure and localization to nucleolar and nuclear vicinities is fully dependent on *XIST* lncRNA.

#### 2.1.3. Genes That Escape from XCI

From 3% to 5% of genes located on mouse Xi chromosome are active, and even more so humans, where up to 30% of genes escape silencing [95,96,97]. Their expression varies depending on species, tissue, cell type, cell cycle, or developmental stage and can contribute to sex differences in gene expression. Genes with the highest escape level (around 75% of Xa counterparts) are localized in pseudoautosomal regions PAR1 and PAR2, which are small regions of homology and meiotic pairing with the male Y chromosome. The rest of the escapees’ genes have an expression level 4 or 5 times lower comparing to Xa counterparts. In humans, those active genes are present in clusters; however, in the mouse genome they are dispersed throughout the whole Xi. In general, the gene is considered to be an active escapee, when its expression level from Xi represents at least 10% of Xa expression. Escaped genes are devoid of the *XIST* lncRNA coating [98]. They are associated with an open chromatin conformation, RNA polymerase II occupancy, active histone marks, and CTCF protein clustered around and YY1 transcription factor bound at promoters [99,100,101]. The transcriptomic study of X-chromosome conducted in different regions of the mouse brain [102] uncovered the existence of genomic regions of actively transcribed lncRNAs, which nonrandomly colocalized with protein-coding genes of escapees. This indicates the possible role of additional lncRNAs in the regulation of X-inactivation escape in the mouse brain.

Genes that escape X-chromosome inactivation may be particularly important in brain function. In humans, the X chromosome bears about 5% of the whole genome, yet it contains about 15% of all known genes associated with intellectual disability. This makes the X chromosome an attractive target in neurobiological research. There are 141 X-linked intellectual disability (XLID) genes, the duplication of which results in some form of mental retardation [103], suggesting their dosage sensitivity. Moreover, detailed analysis of the global expression of X-linked genes showed their higher expression in the brain as compared to other tissues [104]. It seems that the brain is one of the tissues where imbalanced dosage compensation is not well tolerated [105]. The majority of genes that escape silencing on Xi are involved in brain development and have been implicated in XLID syndromes [95]. An example is *KDMC5* coding lysine-specific demethylase 5C (KDM5C, also known as JARID1C or SMCX), the mutations of which cause Claes–Jensen type XLID (CJ-XLID), a rare syndrome accounting for approximately 1–3% of all XLID cases. KDM5C plays a critical role in constraining transcription during neuronal differentiation and maturation by removing from the histone tail active posttranslational modification, H3K4me3/2 [106]. In addition to severe intellectual disability, CJ-XLID is characterized by autistic behavior, hyperreflexia, emotional outbursts, epileptic seizures, and spastic paraplegia [107,108]. Additionally, KDM5C was shown to be involved in psychiatric disorders. Ji et al. [109] found that the *KDM5C* and *XIST* genes are significantly over-expressed in lymphoblastoid cells and postmortem brains of female patients with bipolar disorder or major depression. The authors suggested that over-expressed *XIST* lncRNA may excessively recruit *XIST*-binding complexes and by this impair XCI, or the *XIST* overexpression might be the response compensating inefficient XCI. The subtle alteration of XCI, which is the cause or the consequence of *XIST* over-expression, results in the unbalanced expression of X-linked genes including X inactivation escapees like *KDM5C*, which in turn regulates the expression of other genes involved in proper brain function. Interestingly, among genes differentially expressed in women, in those who developed co-morbid chronic musculoskeletal pain (CMSP) and posttraumatic stress syndrome (PTSS) in consequence of motor vehicle collision, 40 genes were localized on the X chromosome [110]. Moreover, many of them were previously described as Xi escapees, and their increased expression correlated with the reduction of *XIST* lncRNA, possible partial derepression of Xi, and later development of CMSP and PTSS.

Another line of evidence of the escapees’ importance in brain function comes from the data on sex chromosome aneuploidies, which occur in 1/500 birth in humans. Human karyotype can be described as 46,XX or 46,XY, standing for 46 autosomes ad two sex chromosomes. As many as 99% of embryos with 45,X karyotype die in utero due to defects in placenta development. The surviving 1% express severe phenotypes of Turner syndrome (TS), including ovarian dysgenesis, short stature, webbed neck, and other physical abnormalities accompanied by deficits in visuospatial reasoning skills, executive function, social cognition, and aberrant brain structure [111]. So far, few specific escapee genes have been found to play a direct role in TS pathology. For example, the SHOX transcription factor, the gene of which is located in the PAR region of chromosome X, is responsible for the short stature of TS patients [112]. It is believed that lower expression of escapee genes like *KDMC5* and *NLGN4X* may be engaged in the specific neurocognitive profile of TS [113]. Approximately half of TS patients express 45,X karyotype, and the remaining cases involve sex chromosome mosaicism (45,X/46,XX) and/or structural abnormalities of the second X chromosome, including the ring X chromosome [114]. The ring X chromosome often remains active due to the absence of the *XIST* locus. It has been shown that the presence of the ring X chromosome was related to cognitive impairments, low IQ scores, and mental retardation, and its size correlated with the severity of neuro-cognitive disabilities. Not only deficiency but also the excess of X chromosomes like in Klinefelter syndrome (KS) has unfavorable consequences for neuro-cognitive functions. A total of 80–90% of KS patients have classical 47,XXY karyotype, and the rest show even more sex chromosomes (48, XXXY or XXYY, and 49, XXXXY). The supernumerary X chromosomes undergo silencing, except for the escapees’ genes. Their altered expression can be involved in the KS phenotype, namely tall stature, gynecomastia, hypogonadism, absent spermatogenesis, changes in brain structures, motor, verbal, and cognition deficits as well as psychiatric disorders from major depression to bipolar disorder to schizophrenia, which are all more frequently diagnosed in KS individuals as compared to the general population [[115]，[116],[117]]. All these data clearly show the significant role of *XIST* and other lncRNAs in the regulation of Xi structure, localization, silencing, and in consequence transcriptional activity of X-linked genes, the alternation of which has deleterious consequences for neuro-cognitive functions.

### 2.2. NEAT1 and Paraspeckles

Nuclear Enriched Autosomal Transcript 1/Nuclear Paraspeckle Assembly Transcript 1 (*NEAT1*) is one of the key architectural lncRNAs and is responsible for paraspeckle formation. This lncRNA is transcribed by RNA pol II [118], and its expression results in the formation of two transcripts: shorter, polyadenylated *NEAT1_1* (3,7 kb in humans) and longer *NEAT1_2* (23 kb in humans) [119] The longer transcript is crucial for paraspeckle assembly as it provides a scaffold for paraspeckle protein binding, and its knockdown results in paraspeckle disintegration [118,120,121,122]. *NEAT1_1* is also found in paraspeckles; however, its role is dispensable for their formation or maintenance [119,123]. On average particular paraspeckles are formed by 53 *NEAT1_2* molecules and 6.5 *NEAT1_1* molecules [124]. More than 40 proteins participate in the paraspeckle formation, most of which belong to the mammalian family *Drosophila melanogaster* behavior human splicing (DBHS) proteins [125]. Several members of this family, like NONO (p54nrb), splicing factor proline/glutamine-rich (SFPQ), or FUS are crucial for paraspeckle assembly [126]. Proteins forming paraspeckles usually possess disordered prion-like domains (PDL), which mediate paraspeckle formation via liquid–liquid phase separation [127]. However, to phase separate, these proteins need the locally-increased concentration of scaffold *NEAT1* RNA [128]. The interaction between *NEAT1_2* and paraspeckle proteins is strong and stable as chemical extraction of *NEAT1_2* from paraspeckles requires harsh conditions [124].

Paraspeckles form in a *NEAT1_2* transcription-dependent manner near the *NEAT1* locus and spread into the nucleoplasm [118,119]. The active transcription by RNA Pol II is necessary for paraspeckle formation and maintenance [129]. Studies in human cell lines have shown that the *NEAT1* promoter contains binding sites for a variety of transcription factors, and several pathways regulating *NEAT1* expression have been described, most of which are connected with stress response [130]. A detailed analysis has shown that the middle part of *NEAT1_2* is crucial for paraspeckle assembly, as it provides a scaffold that binds NONO dimers [131]. Binding of NONO to the mid-portion of *NEAT1_2* initiates oligomerization of other DHSB proteins, which leads to paraspeckle assembly via a phase-separation mechanism [131]. Paraspeckles have a spheroidal core–shell structure with a *NEAT1_2* folded Vi-shape, so that its 3′ and 5′ ends localize in the shell region of the paraspeckle and its middle part occupies the core region. This organization impacts the distribution of paraspeckle proteins. DHSB proteins colocalize with the *NEAT1_2* mid part in the core of paraspeckles, while proteins like TARDBP, RBM14, or BRG1 with other SWI/SNF complex proteins colocalize with the *NEAT1* ends [132,133]. The number and length of paraspeckles are directly proportional to the level of *NEAT1_2* transcription [134]. The use of electron microscopy and histone H3 detection showed that paraspeckles are mostly depleted of chromatin [133]. However, it is known that *NEAT1* may bind active chromatin sites and that its localization changes significantly in response to changes in transcription [135]. Moreover, Chakravarty et al. [136], working on prostate cancer cells, showed that *NEAT1_1* can directly interact with promoters of a particular set of genes and induce the active chromatin state favorable for transcription. This is probably mediated via the interaction of *NEAT1* with histone H3 and its modifications H3K9Ac and H3K4me3. This raises the question of whether it fulfills this function independently or as a part of paraspeckles, and consequently, whether paraspeckles are freely diffusing bodies or whether they might bind to the chromatin. Interestingly, a recent study by Grosh et al. has provided evidence that paraspeckles might be tethered to chromatin most probably by forming a triple helix structure [137]. If so, *NEAT1* might affect 3D genome architecture by acting as a molecular bridge between chromatin and paraspeckles [22,135].

Paraspeckles form in numerous cell lines and various tissues; however, it seems that they are not vital for cell viability, at least under normal conditions [123]. Till now, paraspeckles have been proposed to (1) be engaged in regulation of transcription, as their presence might affect the availability of transcription factors like NONO and SFPQ [134,138], (2) the regulation of translation, by sequestrating A-to-I hyper-edited RNAs [132], (3) to act as an miRNA sponge, which is mediated by the ability of *NEAT1* to bind microRNAs. Consequently, paraspeckles are considered as a “buffer” for various miRNAs, thus affecting the transcription of numerous genes targeted by these miRNAs [139,140,141]. Importantly, several data show that paraspeckle formation is enhanced by stress conditions, e.g., proteasome inhibition [134,138,142,143].

Despite accumulating data, still very little is known about the role of *NEAT1* in the neuronal cell nucleus. The expression of both *NEAT1* isoforms in the brain, as estimated by qPCR, is very low [123,144,145]. When analyzed separately in the cortex, cerebellum, and spinal cord *NEAT1_1* showed modest expression, while *NEAT1_2* was hardly detectable [144]. Similar data were obtained when *NEAT1* levels were analyzed using FISH in particular parts of the nervous system [144,146]. Analysis of different brain structures confirmed that the dominant form expressed is *NEAT1_1* [146]. Furthermore, *NEAT1_2* could not be detected in the spinal cord motor neurons. In tissue samples from young and old mice, the only detectable isoform of *NEAT1* was *NEAT1_1*. In contrast to neurons, single-cell RNA-seq data showed that *NEAT1* expression in glial cells is relatively high [144].

Until recently, it has been thought that loss of *NEAT1* does not have any significant impact on mouse viability or neuronal functions, though it does impact the development of the corpus luteum [147]. However, a recent study by Kukharsy et al. shed new light on this topic [146]. The group provided evidence that *NEAT1* knockout mice show an exaggerated response to physiological stress. In these mice, neurons were hyperexcitable, most probably due to the up-regulated voltage-gated Na+ influx [146]. Moreover, they showed that loss of *NEAT1* in the brain leads to the perturbations in alternative splicing, which particularly affects genes related to RNA metabolism, synaptic functions, and neurological diseases [146]. Interestingly, it has been also shown that *NEAT1* knockdown alters the expression of genes serving neuroprotective roles, and downregulation of key paraspeckle proteins like NONO or SFPQ negatively affects cell viability [148]. These data suggest that *NEAT1* might be important for proper neuronal functioning. However, the analysis of the role of *NEAT1* in neuronal activity has given contradicting results. Lipovich et al. [149] provide evidence for activity-dependent *NEAT1* function in the human neocortex. Using brain samples from human patients suffering from epilepsy, they identified *NEAT1* as one of the lncRNAs upregulated in brain areas of increased activity. Next, using the SH-SY5Y neuroblastoma cell line, they confirmed these results showing that under chronic depolarization, *NEAT1* expression was upregulated within 4 h. Those results were, however, contradicted by recent data obtained by Butler et al. [145]. Using Na2 cells and primary hippocampal pyramidal neurons, the group showed that *NEAT1* levels decrease after KCl stimulation. The results were confirmed in the animal model. After 1 h of contextual fear conditioning, mice had significantly reduced *NEAT1* levels in the dorsal hippocampus. The reduction of *NEAT1* levels was coupled with an increase in cFos expression both in cultured cells and in mouse brains. Importantly, in adult mice, changes in *NEAT1* level were correlated with perturbations in hippocampus-dependent memory formation. The researchers proposed the mechanism according to which *NEAT1* affects the chromatin accessibility by interacting with EHMT2 methyltransferase and adding methyl marks to H3K9 [145]. Similar results concerning NEAT1 expression after neuronal activation were obtained by Barry et al. [150]. This group also proposed a possible mechanism by which *NEAT1* might affect neuronal excitability [150]. They showed that *NEAT1* expression is inversely correlated with the expression of ion channel genes. Moreover, they provided evidence that it can also directly interact with potassium channel interacting proteins. These proteins are one of the key components engaged in regulating neuronal excitability [151]. Residing in the nucleus, *NEAT1* may act as a scaffold that upon neuronal activation, releases modulatory proteins to fine-tune the excitatory response [150]. Transient downregulation of *NEAT1* upon neuronal activation might induce the release of potassium channel proteins from the nucleus into the cytosol. This data shows that *NEAT1* is involved in processes crucial for the proper functioning of neurons. Additional evidence that NEAT1 is important for these cells comes from the analysis of neuronal aging and pathological conditions. Accumulating data show that *NEAT1* can be dysregulated in aging brains and several neuronal diseases, such as Huntington’s disease, amyotrophic lateral sclerosis (ALS), or frontotemporal dementia (FTD) [145,148,152,153,154]. However, in different diseases, *NEAT1* might be either up- or down-regulated. The imbalance in *NEAT1* expression is often accompanied by mutations in various paraspeckle proteins like TDP-43 or FUS [144,154,155,156]. One of the best-characterized examples is ALS and FTD. It has been shown that in motor neurons of ALS and FTD patients as well as mouse models of these diseases, *NEAT1* becomes upregulated and is excessively bound to TDP-43, which contributes to decreased neuronal cell viability [154,157]. Interestingly, the authors checked whether *NEAT1* knockdown could reverse this effect, and it appeared to have a similarly negative impact on neuronal viability as upregulation. This led them to the conclusion that any imbalance in *NEAT1* expression in motor neurons might have negative impact on the cell viability [154] Similarly, increased expression of *NEAT1* has been observed in a mouse model of Huntington’s disease and ataxia types 1,2, and 7 [148]. On the other hand, analysis of a mouse model of the AD dataset from the National Center for Biotechnology Information (NCBI) database showed that *NEAT1* is downregulated in the hippocampus at the early stages of AD [158] These data highlight that although a very low *NEAT1* expression level is important for neurons, its imbalance might contribute to the development of a pathological phenotype.

It seems that at least some of these diseases are accompanied by de novo paraspeckle formation in the neuronal cell nucleus. Probably the first evidence for the possible formation of paraspeckles in neuronal cells was shown by Nishimoto et al., who analyzed paraspeckles in amyotrophic lateral sclerosis (ALS) tissue samples from mice and humans [144]. Shelkovnikova et al. further showed that while normal neurons lack paraspeckles, they are present in ~40% of motor neurons of patients with ALS. They also showed that loss of TDP-43 protein is sufficient to stimulate paraspeckle formation. The probable underlying mechanism involves compromised miRNA biogenesis or activation of the dsRNA response [159]. These results point to the possible role of paraspeckles in regulating the miRNA pathway, which is in line with the evidence obtained on the cell lines [141]. Interestingly, also in the Huntington’s disease model, enlarged paraspeckles have been observed [148], which suggests that paraspeckles might play a neuroprotective role in stress conditions. On the other hand, it remains unclear how the nucleation of paraspeckles is regulated. If active transcription of *NEAT1* is necessary for paraspeckles to form, then what level of this transcription will result in the appearance of paraspeckles?

Together this data show that though barely detectable in the neuronal cell nucleus, *NEAT1* plays an important role in neuronal activation, the fundamental process in the neuronal cell. It seems that *NEAT1* acts independently of paraspeckles, as available data show that normal neurons lack these nuclear domains. This suggests that at basal conditions (silent neurons), the main role is played by the *NEAT1_1* transcript. However, under stress conditions, the cell switches to longer *NEAT1* isoform synthesis. The appearance of *NEAT1_2* leads to nucleation of paraspeckles, which help the cell to restore balance.

### 2.3. MALAT1 and Nuclear Speckles

*MALAT1* (metastasis-associated lung adenocarcinoma transcript 1) was originally identified in a screen for RNAs, the levels of which were altered in early-stage lung cancers, and was found to be upregulated in tumors with a high ability to metastasize [160] Unlike most of the known lncRNAs, *MALAT1* has high sequence conservation in vertebrates [161]. The *MALAT1* gene is transcribed by RNA pol II, and its transcription gives rise to ~7 kb lncRNA. It is highly enriched in nuclear bodies called splicing speckles, where it might play a role in regulating the protein composition of these nuclear bodies [162].

Splicing speckles are liquid-like nuclear bodies enriched in serine/arginine (SR) splicing factors and SR-like proteins. The human interphase nucleus contains 20–40 speckles, ranging in size from 0.5 to several micrometers in diameter [163]. Both speckle number and shape respond dynamically to changes in transcriptional activity [164,165,166]. Speckles have complex protein compositions including the aforementioned splicing factors, transcription factors, 3′end RNA processing factors, proteins engaged in mRNA export, and enzymes regulating splicing machinery [163]. Their composition suggests that they may orchestrate crucial steps of transcription [167]. Moreover within speckles, apart from *MALAT1*, different types of RNAs can be found, including pre-mRNAs and snoRNAs [167,168,169]. Their complex composition is reflected in their organization, in which SR and SR-like proteins form the core of the speckle, and snRNAs and *MALAT1* are enriched in the speckle periphery [170].

It has been shown that actively transcribed genes can associate with nuclear speckles [171,172,173,174,175,176,177,178,179], and this association increases their transcription rate [180]. The development of novel 3D interaction analysis techniques like split-pool recognition of interactions by tag extension (SPRITE), or tyramide signal amplification (TSA)-seq has revealed that gene-speckle association is a common event. This leads to the suggestion that splicing speckles play an important role in shaping 3D chromatin architecture by acting as hubs, which recruit active genes and bring in close spatial proximity genes laying on different chromosomes [181,182]. Interestingly, the transcription of speckle-associated genes and the accumulation of RNA in speckles affects their size [170].

As a component of splicing speckles, *MALAT1* has been shown to play various roles. Due to its ability to affect the levels of speckle components and interact with different speckle proteins, it participates in regulating speckle number and organization [162,170,183]. It can serve as a “sponge” for splicing factors, thus modulating the recruitment of SR splicing factors to an active transcription site [36,38] and regulating spliceosome composition within speckles, which affects alternative splicing. By its ability to bind miRNAs, *MALAT1* can also regulate their availability for the downstream processes [36]. Though the detailed mechanism is unknown, it seems possible that *MALAT1* regulates the levels of miRNA while being associated with nuclear speckles, as miRNA processing machinery was found in splicing speckles [184]. Interestingly, though engaged in multiple speckle-related processes, *MALAT1* knockout has no obvious effect on speckle formation and maintenance [185,186]. Thus, it might be dispensable for their organization. On the contrary, a recent study by Nguyen et al. [187] has shown that deletion of the SINEB1 element from *MALAT1* significantly affected speckle morphology. This deletion perturbed *MALAT1*-protein interactions, indicating that they might be important in speckle stabilization. It also resulted in the mislocalization of *MALAT1* to the cytoplasm, where it “hijacked” speckle protein TDP-43, leading to the formation of the cytotoxic TDP-43 inclusions. These results suggested that *MALAT1* might be also important for proteostasis [187]. *MALAT1* binds numerous active gene sites and has been shown to control gene expression in *cis* [186,188] and *trans* [135]. Due to its ability to interact directly with chromatin, *MALAT1* might tether splicing speckles to chromatin fiber [135]. Accordingly, *MALAT1*, together with another lncRNA, *TUG1*, has been shown to affect the 3D nuclear organization. Both lncRNAs act to relocate the growth control genes from the repressing environment of polycomb bodies to the transcriptionally active milieu of splicing speckles by interacting with Polycomb 2 (Pc2) protein present on gene promoters [189]. In this way, both lncRNAs and Pc2 provide a mechanism that determines the relative positioning of particular genes with respect to subnuclear architectural structures [189].

Little is known about the role of *MALAT1* in neuronal cell nuclei. *MALAT1*-enriched nuclear speckles are a common feature of the neuronal nucleus [190], where they form a dynamic domain that reorganizes in response to external cues [191]. Considering speckle’s role as conductor of transcription, they should be particularly important in neurons, which dynamically change their transcriptome in response to stimulation. Accordingly, *MALAT1* is expressed in neurons at high levels [38,145,186]. It has been shown that downregulation of *MALAT1* in primary cortical neurons results in their increased excitability [192]. Results obtained by Lipovich et al. showed that *MALAT1* is upregulated in high-activity areas of the neocortex in patients with epilepsy, and its expression pattern suggested that it is engaged in the regulation of activity-dependent genes [149]. Moreover, knock-out of *MALAT1* in cultured hippocampal neurons led to decreased synaptic density [38], which suggests that it also plays a role in regulating genes involved in synapse formation/maintenance. Nonetheless, despite its obvious role in neurons, it was shown that its knockout does not affect mouse pre- and postnatal development [186]. However, it is tempting to speculate that it might affect the behavior of these animals, but unfortunately such data are lacking.

Several papers point to the importance of the regulatory role of *MALAT1* in neurons due to its ability to sponge miRNAs. The interaction of *MALAT1* with miR-30 is involved in the regulation of neurite outgrowth in hippocampal neurons [193]. It also plays a protective role in ischemic stroke. Both in cultured primary cerebral cortex neurons and the mouse ischemic stroke model, *MALAT1* was downregulated, which led to an increase in miR-30a miRNA and, in consequence, inhibition of autophagic pathways [194]. *MALAT1* levels were significantly elevated in the hippocampi of rats with epilepsy, and its downregulation exerted a protective effect on neuronal survival [195]. Similarly, *MALAT1* is upregulated in the PD mouse model, and its ability to sponge miR-124 was one of the causes of neuronal apoptosis [196]. On the other hand, studies on the AD rat model showed that the level of *MALAT1* was significantly decreased [197]. Its overexpression prevented neuronal apoptosis and promoted neurite outgrowth [198]. Thus, it seems that in different neuronal diseases, *MALAT1* might play contradictory roles, though at least some of them, e.g., *MALAT1*–miRNA interaction, might have a neuroprotective function [194,197].

### 2.4. Gomafu and Gomafu Bodies

The study of Sone and colleagues [40] showed that a novel lncRNA, *GOMAFU*, also known as *MIAT* (myocardial infarction associated transcript) or *RNCR2* (retinal non-coding RNA 2), was highly expressed in differentiating neural progenitor cells. In the mature brain, *GOMAFU* was detected in a subset of postmitotic neurons including retinal ganglion and amacrine cells of the retina, and pyramidal cells in the cerebral cortex layer V and CA1 region of the hippocampus. It formed distinct nuclear bodies but yet failed to colocalize with markers of known nuclear domains like PML and Cajal bodies, nuclear speckles, paraspeckles, or nucleolus. Therefore, the authors suggested that *GOMAFU* forms a new kind of nuclear domain embedded in the nuclear matrix. Unlike speckles and paraspeckles, *GOMAFU* body formation and maintenance were not dependent on transcriptional activity. Later studies have suggested that *GOMAFU* is involved in the regulation of several neuronal processes like retinal cell fate specification [199], neuronal and glial cell differentiation [200,201], the survival of newborn neurons in corticogenesis [202], and cognitive decline in aging [153,203]. Moreover, differential screening showed its higher expression in the nucleus accumbens of heroin- and cocaine-addicted users [204]. Knockout mice completely devoid of *GOMAFU* expression did not present any developmental drawbacks but exhibited hyperactive behavior and sensitivity to psychostimulants, which was associated with an increased extracellular concentration of dopamine in the same brain region known to be involved in addiction [205]. Transcriptomic analysis showed changed expression of only 19 transcripts; however, among them were several genes involved in important neuronal functions. For example, KO mice had decreased expression of CEBPB (CCAAT-enhancer-binding protein), a transcription factor implicated in memory consolidation in the hippocampus. Interestingly the expression of *GOMAFU* is downregulated in cortical neurons in the response to activation. Additionally, its decreased level was found in the cortical gray matter of the superior temporal gyrus of schizophrenia patients [206]. The loss-of-function mutations of *GOMAFU* lead to the alternative splicing patterns in transcripts of the synaptic plasticity-related genes. As *GOMAFU* was shown to bind splicing factors QKI and SRSF1 (serine/arginine-rich splicing factor 1), it is proposed to regulate alternative splicing of transcripts involved in neuronal activation and pathology of schizophrenia. Studies of the Nakagawa group [205,206] seem to confirm this scenario, as they demonstrated *GOMAFU* interaction with two other well-recognized splicing factors, SF1 and CELF3. Both proteins assembled into a novel nuclear domain, called by the authors CS bodies, which were sensitive to transcription inhibition and RNAse treatment. Surprisingly, despite the dependence between the expression of CELF3 protein and transcription of *GOMAFU*, CS bodies did not contain *GOMAFU* RNA itself. All the aforementioned publications point out the activity-dependent function of *GOMAFU* in the postmitotic neurons. Even though the exact molecular pathways are poorly recognized, this abundant lncRNA, which forms a unique nuclear domain in a specific subset of mature neurons, plays an important role in the regulation of genes involved in neuronal plasticity.

## 3. Perspective

In summary, we have described how lncRNAs shape the neuronal nucleus architecture and stand behind its dynamic plasticity. In the context of neuron-related diseases, we discuss how lncRNA imbalances might contribute to the pathological phenotype by affecting the organization of subnuclear domains. It is not yet clear how nuclear architecture translates to gene expression, but accumulating data point to a link between them.

The significance and exact molecular mechanisms underlying lncRNA function in neuronal plasticity are still far from understood. Altogether, these studies indicate that lncRNAs and RNA binding proteins reside in specific nuclear compartments that dynamically respond to neuronal stimulation.

## Figures and Tables

**Figure 1 ijms-22-03478-f001:**
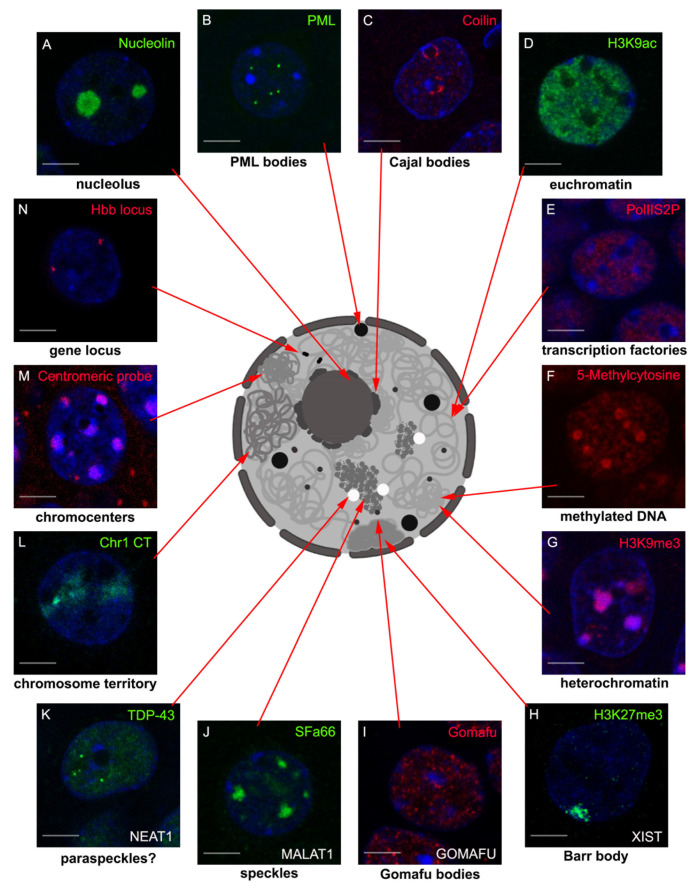
Functional and structural domains of the neuronal cell nucleus. To emphasize the complexity of the neuronal nucleus, different functional and structural assemblies were stained in rodent hippocampal neurons using either antibody recognizing marker proteins or FISH probes complementary to specific sequences. The intricate architecture of the neuronal nucleus consists of nuclear bodies like the nucleolus (**A**, [30]), and PML (**B**, [16]) and Cajal bodies (**C**, [17]); regions of actively transcribed genes marked by the acetylated histones (**D**) and activated RNA polymerase II (**E**); and regions of silenced chromatin characterized by methylated DNA (**F**) and methylated histone H3 (**G**). It contains also nuclear domains like Barr bodies (**H**), Gomafu bodies (**I**), speckles (**J**), and paraspeckles (**K**), the assembly and function of which fully rely on lncRNAs (highlighted in white). The genome components like gene loci (**L**), chromosome territories (**M**), and repetitive sequences in chromocenters (**N**) are also shown. Scale bar is 5 µm.

## Data Availability

Not applicable.

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
