# Peer review of "Long Noncoding RNAs—Crucial Players Organizing the Landscape of the Neuronal Nucleus"

_ijms, 2021, doi:10.3390/ijms22073478_

Round 1

Reviewer 1 Report

The review by Sas-Nowosielska Hand Magalska A provides a comprehensive summary of some important lncRNAs function in nucleus of the neuronal cells. The text is up to date with the published peer-reviewed literature and gives a mechanistic explanation of several lncRNAs vital in neuronal nuclear metabolism. The document is easy to follow. However, below are some suggestions that may further improve the review.

  1. While the information in text is very thorough, it will be convenient to the readers if the authors include a graphical summary of the discussed lncRNA function. This will be consistent with the increasing trend adapted by many scientific journals to increase the impact of the research work.
  2. The two figures included in the manuscript are helpful, however some of the aspects are not well-addressed in the main text or figure legends. For example, Cajal and PML bodies in fig.1. The authors may expand on this in the figure legend. Further, fig. 2 does not add any significant value to the discussion to the main text. It seems to be an accessory figure.
  3. While authors comprehensively discuss the role of Xist, it will be a good perspective to include roX1 and roX2 lncRNAs that are involved X-chromosome activation in Drosophila. These have also been implicated as regulators of chromatin organization.
  4. The authors may also briefly discuss the role of TSIX transcribed from Xa that represses early Xist function.
  5. Role of Meg3 and other Dlk1and Dio3 locus derived lncRNAs in epigenetic regulators of progenitor and some Hox genes in mouse neurons. (work by Yen YP et. al eLife 2018 and others).
  6. One recent finding suggests the role of Malat1 in TDP-43 proteostasis. Particularly in stimulating assembly of cytotoxic nuclear inclusions that contribute to neurodegeneration (see Nguyen TM et.al Nucleic Acid Research 2020). The review may benefit from discussing this recently characterized function of Malat1.
  7. Information in lines 138-140 and 141-144 is difficult to follow due to a single long sentence discussing multiple findings. Please break it into smaller sentences and/or use punctuation so that it is easy to follow.
  8. Please check the information about number of molecules for each Neat1 isoform discussed in line 231. Read reference 118.
  9. There are typos, missing words, and misspellings in lines 50, 70, 133, 156, 158, 171, 176, 253, 266 and 323. There may be more.

Reviewer 2 Report

In this review Sas-Nowosielska and Magalska consider molecular and physiological functions of several nuclear long noncoding RNAs in neuronal cells. They focus on describing the regulatory role of Xist RNA, directing X-chromosome inactivation, as well as on transcripts involved in the formation of intranuclear compartments through liquid-liquid phase separation. The authors pay special attention to data on the connections of these RNAs with diseases of the nervous system. The review is clearly written, covers most of the latest works, and is of great interest to researchers in this field.

Some minor remarks should be addressed prior to publication:

  1. page 2, lines 67-69. The authors indicate that “The recent analysis of 500 public RNA-seq datasets from multiple tissues and species showed that in mammals and birds the transcriptional differences between sex chromosomes and autosomes is approximately 0.5 “. Firstly, the reference is missing here. Second, it is not clear how this information is related to the phenomenon of dosage compensation, the meaning of which is precisely in the equalization the expression of genes between sex chromosomes of different biological sexes but not between sex chromosomes and autosomes. Therefore, both X-inactivation in females (mammals) and X upregulation in males (Drosophila) are usually considered as different types of dosage compensation mechanisms in contrast to what the authors state in this paragraph.
  2. page 5, lines 192-194. It is reasonable to assume that Xist overexpression will lead to increased silencing of X-chromosomal genes that normally escape this repression. How, then, can one explain the cited work (Ji et al. [103]), where simultaneous upregulation of both KDM5C and XIST genes was observed?
  3. In the Introduction the authors note that “Here we will focus on those lncRNAs that directly affect the organisation of the neuronal nucleus.” However, this is not true in the case of Malat1. I think it would be good for the readers if the authors more clearly formulate in the introduction why they consider these particular lncRNAs and do not mention other well-known types, such as nucleolar lncRNAs, enhancer RNAs, other miRNA sponges, HOTAIR etc.
